# Mipsagargin: The Beginning—Not the End—of Thapsigargin Prodrug-Based Cancer Therapeutics

**DOI:** 10.3390/molecules26247469

**Published:** 2021-12-09

**Authors:** John T. Isaacs, William Nathaniel Brennen, Søren Brøgger Christensen, Samuel R. Denmeade

**Affiliations:** 1Department of Oncology, Sidney Kimmel Comprehensive Cancer Center (SKCCC), Johns Hopkins University School of Medicine, Baltimore, MD 21287, USA; wbrenne2@jhmi.edu (W.N.B.); DENMESA@jhmi.edu (S.R.D.); 2Department of Pharmacology and Molecular Science, Johns Hopkins University School of Medicine, Baltimore, MD 21287, USA; 3Department of Urology, James Buchanan Brady Urological Institute, Johns Hopkins University School of Medicine, Baltimore, MD 21287, USA; 4Department of Drug Design and Pharmacology, University of Copenhagen, DK-2100 Copenhagen, Denmark; soren.christensen@sund.ku.dk

**Keywords:** Thapsigargin, Mipsagargin, tissue-specific proteases, targeted prodrugs, SERCA, *Thapsia garganica*, apoptosis, calcium homeostasis

## Abstract

Søren Brøgger Christensen isolated and characterized the cell-penetrant sesquiterpene lactone Thapsigargin (TG) from the fruit *Thapsia garganica.* In the late 1980s/early 1990s, TG was supplied to multiple independent and collaborative groups. Using this TG, studies documented with a large variety of mammalian cell types that TG rapidly (i.e., within seconds to a minute) penetrates cells, resulting in an essentially irreversible binding and inhibiting (IC_50_~10 nM) of SERCA 2b calcium uptake pumps. If exposure to 50–100 nM TG is sustained for >24–48 h, prostate cancer cells undergo apoptotic death. TG-induced death requires changes in the cytoplasmic Ca^2+^, initiating a calmodulin/calcineurin/calpain-dependent signaling cascade that involves BAD-dependent opening of the mitochondrial permeability transition pore (MPTP); this releases cytochrome C into the cytoplasm, activating caspases and nucleases. Chemically unmodified TG has no therapeutic index and is poorly water soluble. A TG analog, in which the 8-acyl groups is replaced with the 12-aminododecanoyl group, afforded 12-ADT, retaining an EC_50_ for killing of <100 nM. Conjugation of 12-ADT to a series of 5–8 amino acid peptides was engineered so that they are efficiently hydrolyzed by only one of a series of proteases [e.g., KLK3 (also known as Prostate Specific Antigen); KLK2 (also known as hK2); Fibroblast Activation Protein Protease (FAP); or Folh1 (also known as Prostate Specific Membrane Antigen)]. The obtained conjugates have increased water solubility for systemic delivery in the blood and prevent cell penetrance and, thus, killing until the TG-prodrug is hydrolyzed by the targeting protease in the vicinity of the cancer cells. We summarize the preclinical validation of each of these TG-prodrugs with special attention to the PSMA TG-prodrug, Mipsagargin, which is in phase II clinical testing.

## 1. Introduction

Over the last decade, there has been a significant improvement in the survival of metastatic prostate cancer (mPCa) patients by the combination of luteinizing hormone-releasing hormone (LHRH) analog-dependent androgen deprivation therapy (ADT) with sequential Androgen Receptor-Axis-Targeted therapies (ARAT) such as Abiraterone/Enzalutamide/Apalutamide/Darolutamide or with taxane-based chemotherapy [1,2,3]. Likewise, Poly (ADP-ribose) polymerase (PARP) inhibitors have antitumor activity against mPCa with DNA damage response (DDR) alterations in genes involved directly or indirectly in homologous recombination repair (HRR) [4]. Unfortunately, despite these major advancements, such increases in survival are measured in months not years [3]. These advancements clearly support that further improvement in survival requires simultaneous or sequential combinations with additional drugs targeting the vulnerabilities of mPCa. As a rationale for our approach to such effective drug development, an understanding of the role of the sarcoplasmic/endoplasmic reticulum calcium transport ATPase (SERCA) pump in regulating intracellular free calcium (Ca^2+^) and the unfolded protein response (UPR) in cell death of prostate normal and malignant cells is required.

## 2. Role of Calcium in the Programmed Cell Death of Normal and Malignant Prostatic Epithelial Cells

In the normal adult prostate, the epithelial cells are continuously turning over with time. In this self-renewing condition, the rate of prostatic cell death is balanced by an equal rate of prostatic cell proliferation such that neither involution nor overgrowth of the gland normally occurs with time. If an adult male is castrated, the serum testosterone level rapidly decreases to below a critical value [5,6]. As a result, the prostate rapidly involutes. This involution is due to a major loss in the epithelial, not the stromal, compartment of the prostate [5,6]. The chronic requirement for androgen by the epithelial compartment is because androgens regulate the total prostatic epithelial cell number by affecting both the rate of glandular proliferation and death of the glandular cells. Androgen does this by chronically stimulating the rate of cell proliferation (i.e., agonistic ability of androgen) while simultaneously inhibiting the rate of cell death (i.e., antagonistic ability of androgen) of the prostatic glandular epithelium [5]. These androgenic responses are transduced by Androgen Receptor (AR)-initiated transcriptional signaling [7]. Androgen supplied via the circulation binds to the AR within prostate stromal cells and initiates ligand-stimulated AR-dependent transcription of specific target genes within prostate stromal cells, resulting in their production and secretion of a series of peptide growth factors known as “andromedins”, which include IGF-1, EGF, FGF7 (KGF), and FGF10. These stromal-derived paracrine andromedins diffuse across the basement membrane into the epithelial compartment where they bind to their respective cognate receptors, initiating cell signaling cascades that stimulate proliferation and inhibit cell death [5,7].

Studies in the late 1980s by Natasha Kyprianou, while a post-doctoral fellow in the Isaacs laboratory, documented that androgen deprivation (AD) results in the loss of ligand-stimulated AR-dependent transcription of the paracrine andromedins coupled with gain of expression of cell death genes whose transcription is normally ligand-dependent AR-repressed. These AD-induced transcriptional changes initiate a series of energy-dependent discrete biochemical events that lead to a cessation of cell proliferation and the activation of programmed (i.e., apoptotic) death of these androgen-dependent prostatic epithelial cells [6]. By the early 1990s, the Isaacs laboratory documented that besides the loss of andromedin expression, this apoptotic death is initiated by the simultaneous loss of ligand-occupied AR-dependent transcriptional repression in both prostate epithelial and stromal cells. This repression results in enhanced expression of TGF-β receptors in the prostatic epithelial cells coupled with enhanced expression and secretion of TGF-β_1_ ligand by prostate stromal cells after androgen ablation [8,9,10,11,12]. These AD-induced changes activate a TGF-β receptor signaling apoptotic cascade, resulting in an endoplasmic reticulum (ER) stress-induced activation of the Unfolded Protein Response (UPR), as documented by transcription and translational upregulation of GRP-78 (also known as BiP or HSP5a) and Calmodulin (CaM) [13,14]. This results in a delayed (i.e., >24–40 h) sustained μM elevation of cytoplasmic Ca^2+^ within the prostate epithelial cells, activating Ca^2+^/ Mg^2+^-dependent endonuclease present within the cell’s nucleus, which double-strand fragments the genomic DNA, thus irreversibly committing the cell to apoptotic death [8,9,10,11,12,13,14,15,16].

We further documented that during transformation of androgen-responsive normal prostatic epithelial to malignant cancer cells, a shift in the AR axis from a stromal cell-dependent paracrine pathway to a cancer cell autonomous autocrine-dependent pathways occurs such that now ligand-dependent AR signaling is oncogenic, driving both malignant cell survival as well as proliferation [16,17,18]. Due to this gain of oncogenic AR function, prostate cancers (PCa) initially are nearly universally responsive to androgen deprivation therapy (ADT), since this induces an autocrine TGFβ-stimulated apoptotic death of such malignant ligand/AR (A^+^/AR^+^)-dependent PCa cells [7,19]. Unfortunately, such A^+^/AR^+^ PCa cells progress during ADT to become independent of androgen ligand, but still require AR signaling. Thus, these (A^−^/AR^+^) PCa cells relapse, since ADT does not induce their apoptotic cell death [7,20,21].

In 1990, Paula Martikainen from the Department of Pathology, Turku University, Finland joined the Isaacs laboratory as a visiting fellow. Importantly, in collaboration with Natasha Kyprianou, they documented that these ligand-independent A^−^/AR^+^ PCa cells retain the Ca^2+^-dependent cellular machinery to undergo apoptosis as documented by the induction of apoptotic death by treatment of such AD-resistant prostate cancer cells with the calcium ionophore, ionomycin, which depletes Ca^2+^ within the ER and raises the cytoplasmic Ca^2+^ [22]. This requires a concentration of ionomycin of at least 10μM [22]. Unfortunately, at this concentration, ionomycin also kills normal cells and, thus, there is no therapeutic index for the use of ionomycin clinically.

## 3. Søren Brøgger Christensen and Thapsigargin Early Studies

Based upon the validation that ionomycin-induced disruption of the intracellular Ca^2+^ pools kills prostate cancer cells, we searched the literature for alternative more potent agents for such induction. This identified Søren Brøgger Christensen as the pharmacognosist who, from the fruits and roots of the *Thapsia garganica* umbelliferous plant, first isolated, chemically characterized, and named the highly lipophilic, cell-penetrant sesquiterpene-lactone Thapsigargin (Figure 1) [23]. (For a more full history of this discovery, see reference [24].)

To clarify how useful Thapsigargin (TG) became once identified, an understanding of the compartmental calcium gradients and their function in mammalian cells is required (Figure 2). In mammalian cells, there is a major concentration gradient between the 1mM Ca^2+^ in the extracellular fluid and the 50–100 nM Ca^2+^ in the cytoplasm vs. the >500 μM Ca^2+^ in the lumen of the endoplasmic reticulum (ER) [25].

These gradients are functionally important, since there is a constitutive requirement for a high (>5–10 mM) concentration of inorganic phosphate (Pi) in the cytoplasm for metabolism. This requirement for high cytoplasmic mM Pi thus limits Ca^2+^ to <1μM to prevent calcium-phosphate precipitation (i.e., precipitation occurs when ((Ca^2+^) × (Pi) >8 mM^2^)) [27]. Unlike the situation in the cytoplasm, there is only a low concentration of Pi within the lumen of the ER allowing the Ca^2+^ in the luminal ER to be maintained at the elevated level (i.e., >500 μM) required for the proper folding of newly synthesized proteins in the ER. This >5000-fold Ca^2+^ concentration gradient between the ER and cytoplasm is maintained by the ER membrane-bound housekeeping SERCA 2b ATPase calcium uptake pumps, which transfer Ca^2+^ trans-membranous from the cytoplasm into the lumen of the ER. This ER Ca^2+^ gradient can be reversibly discharged, however, by temporary opening of ER calcium efflux channels (e.g., IP_3_ receptors) via their binding of short-lived ligand intermediates (e.g., IP_3_) generated as part of the signaling pathways initiated by activated plasma membrane receptors [26] (Figure 2). This results in a transient elevation in cytoplasmic Ca^2+^, activating not only calcium-dependent kinases and phosphatases, but also enhanced uptake of Ca^2+^ into mitochondria, stimulating ATP production [26].

In the late 1980s/early 1990s, Søren Brøgger Christensen provided TG for study—either directly or through commercial sources to whom he supplied the compound—to multiple independent and collaborative groups around the world. Using his TG, these studies documented using a large variety of mammalian cell types that, due to its high lipophilicity, TG rapidly (i.e., within seconds to a minute) penetrates cells, resulting in its rapid and essentially irreversible binding and inhibition (IC_50_~10 nM) of SERCA 2b calcium uptake pumps in the ER. This inhibition of uptake coupled with spontaneous “leakage” from the ER Ca^2+^ pool down the high concentration gradient induces within <5 min depletion of Ca^2+^ in the ER coupled with elevation in cytoplasmic Ca^2+^ [28,29]. This depletion of Ca^2+^ in the ER activates the Unfolded Protein Response (UPR). Under normal conditions, molecular chaperones such as calreticulin (CRT), Binding-Immunoglobulin Protein (BIP) (also known as Heat Shock 70 kDa Protein 5 or Glucose-Regulated Protein, 78 kDa), calnexin (CNX) and Protein Disulfide Isomerase Family A Member 6 (PDIA6) bind to and buffer the Ca^2+^ in the ER lumen. BiP/CRT and PDIA6 are an especially important Ca^2+^-binding protein as they are involved in sensing misfolded protein accumulation in the ER and, in conjunction with three proximal sensors IRE1, PERK and ATF6, regulate the UPR (Figure 2) [26].

Under non-stressed conditions, Ca^2+^-bound BiP binds to the luminal domains of IRE1, ATF6, and PERK, while PD1A6 co-binds with BiP to IRE1 and CRT co-binds with BiP to ATF6 [26]. Upon accumulation of unfolded proteins in the ER lumen, IRE1 released from BIP/PD1A6 dimerizes to activate its kinase and RNase activities to initiate XBP1 mRNA splicing, thereby creating a potent transcriptional activator. Primary targets that require the IRE1/XBP1 pathway for induction include genes encoding functions in Endoplasmic reticulum (ER)-associated protein degradation (ERAD), needed to remove misfolded proteins. Similarly, ATF6 released from BiP/CRT transits to the Golgi compartment, where cleavage by S1P and S2P proteases yields a cytosolic fragment that migrates to the nucleus to further activate transcription of UPR-responsive genes. Finally, PERK released from BiP dimerizes, autophosphorylates, and phosphorylates eIF2α on Ser 51, leading to general attenuation of translational initiation, thus inhibiting protein synthesis. Paradoxically, eIF2α phosphorylation induces translation of ATF4 mRNA. The PERK/eIF2α/ATF4 regulatory axis also induces expression of antioxidative stress response genes and expression of genes encoding proteins with proapoptotic functions, such as Growth Arrest Additionally, DNA Damage-Inducible Protein (GADD153) (also known as DDIT3 or CHOP). TG inhibition of the SERCA pump not only depletes ER luminal calcium stores inducing UPR—it also triggers the opening of plasma membrane calcium channels in an attempt to refill the ER pool of Ca^2+^ by a process termed store operated Ca^2+^ entry (SOCE). This process is activated by depletion of Ca^2+^ from the ER, which induces stromal interaction molecules 1&2 (STIM1&2) to accumulate at ER–plasma membrane (PM) junctions, where they trap and activate ORAI Calcium Release-Activated Calcium Modulator channels, allowing entrance of Ca^2+^ from the mM pool in the extracellular fluid, which elevates cytoplasmic Ca^2+^ within minutes of TG exposure [30].

## 4. Thapsigargin Drug Development

In 1991, Donald L. Gill’s laboratory in the Department of Biochemistry, University of Maryland School of Medicine reported that when hamster DDT_1_ MF-2 smooth muscle cells are treated with 3μM TG, the rapid (<2 min) SERCA 2b inhibition and resulting ER depletion of Ca^2+^ results in only a transient (i.e., <1 h) elevation of cytoplasmic Ca^2+^ [31]. In contrast to the transient elevation in cytoplasmic Ca^2+^, the TG-induced depletion of the Ca^2+^ in the ER is chronically sustained, resulting in inhibition of both protein and DNA synthesis and, thus, cell proliferation over a one-week period [31]. Significantly, however, while proliferation is inhibited, the DDT_1_ MF-2 smooth muscle cells do not die over this one-week TG treatment period [31].

In 1992, John T. Isaacs presented a seminar at the University of Maryland where he discussed the possibility of using agents targeting calcium regulation as novel therapeutic approaches for castration-resistant prostate cancer. At the seminar was Alison D. Short who, at the time, was a member of the Gill laboratory. Based upon her introduction, the Isaacs and Gill laboratory initiated a collaboration to test the response of prostate cancer cells to TG. The Hopkins portion of these initial studies was performed by Yuzo Furuya, a post-doctoral fellow from the Department of Urology, Chiba, Japan and Per Lundmo a visiting scientist from the Department of Surgery, University of Trondheim, Norway. For these initial studies, TG was obtained from a commercial source and we published that, at 500 nM TG, as predicted, it rapidly depletes ER luminal Ca^2+^ and, thus, inhibits protein synthesis and cell proliferation [32]. What was unexpected was the seminal discovery that in cancer cells, TG treatment induces cell death, which is critically dependent upon an adequate, sustained (i.e., >1 h) elevation in the cytoplasmic Ca^2+^, which we documented for the first time in the literature in 1994 [32]. In this initial report, we documented that following exposure of prostate cancer cells to 500 μM TG, cytoplasmic Ca^2+^ rose rapidly (i.e., within 1 min) from <30 nM to values exceeding >150 nM. During the next 5–10 min, the cytoplasmic Ca^2+^ stabilizes at 100–160 nM, which is maintained for hours, if extracellular Ca^2+^ is maintained. If extracellular Ca^2+^ is lowered to less than 50 nM, the initial TG-induced rise in cytoplasmic Ca^2+^ remains unchanged. However, prolonged elevation of cytoplasmic Ca^2+^ is not observed, and cytoplasmic Ca^2+^ returns to 10–30 nM by an hour. These results suggested that TG induces an initial elevation in cytoplasmic Ca^2+^ as the pool of Ca^2+^ stored in the ER is released, followed by an influx of extracellular Ca^2+^ across the plasma membrane. In subsequent studies, we documented that this TG-induced death of prostate cancer cells produces ER stress, activating UPR as documented by transcriptional and translational upregulation of BiP and CaM [33]. Significantly, while the elevation in cytoplasmic Ca^2+^ is rapid, completion of apoptotic death of the cancer cells requires 24–48 h [32].

In 1993, Samuel R. Denmeade joined the Isaacs laboratory as a Medical Oncology fellow focusing on drug development for the prostate. As will be discussed below, he led the efforts in our group to identify prostate-specific proteases and their restrictive substrates for the development of our prodrug approaches. In 1998, Bertrand Tombal from the Catholic University of Louvain, Brussels, Belgium joined our group at Hopkins as a post-doctoral fellow. Drs. Tombal and Denmeade collaborated on studies to identify the molecular pathway by which TG induces the death of prostate cancer cells. As part of these studies to evaluate the role of increased cytoplasmic Ca^2+^ in TG-induced death of prostate cancer cells occurring over 24–48 h, they developed a method for accurately and simultaneously measuring the kinetics of changes in cytoplasmic Ca^2+^ and morphological changes characteristic of apoptosis longitudinally in individual cells over several days following exposure to TG [34]. Their approach was possible based on the introduction of the tetracarboxylate fluorescent calcium indicators Indo-1 and fura-2 by Tsien and colleagues. Passive loading of the acetoxymethyl (AM) ester derivatives of these dyes was widely used to measure cytoplasmic Ca^2+^ changes in cells undergoing apoptosis [34]. Such passive loading of cells was initially developed, however, for kinetic analysis of acute cytoplasmic Ca^2+^ changes (i.e., seconds, min), but these methods have severe limitations due to dye leakage and compartmentalization into organelles when such kinetic analyses are performed in individual cells, limiting the accuracy of such passive dye loading to longitudinal cytoplasmic Ca^2+^ measurements of <2 h [34]. Post-loading of dye in cells exposed to an apoptosis-inducing agent for several hours is also inaccurate owing to decreased loading efficiency and de-esterification of the probes, resulting in increased production of fluorescent Ca^2+^-insensitive dye species [34]. To overcome these limitations and allowing accurate measurement of the kinetics of cytoplasmic Ca^2+^ changes longitudinally in individual cells undergoing apoptosis, Bertrand Tombal microinjected high molecular weight (i.e., 10,000 or 70,000 MW) fura-dextran fluorescent Ca^2+^ indicators directly into the cytoplasm of cells before initiating treatment. Microinjected fura-dextran is non-toxic to the cells and is resistant to both extracellular leakage and compartmentalization into either the ER or mitochondria [34]. Combining this microinjection method with simultaneous fluorescence ratiometric and phase contrast image analysis allows cytoplasmic Ca^2+^ and morphological changes characteristic of apoptosis (i.e., cell blebbing and eventual fragmentation into membrane-bound apoptotic bodies) simultaneously to be followed longitudinally in individual cells.

Longitudinal kinetic analysis of individual prostate cancer cells demonstrated that TG-induced SERCA inhibition results in rapid (<5 min) depletion and release of the ER Ca^2+^ pool into the cytoplasm. One of the earliest (i.e., within one h) effects of TG-induced inhibition is a 3-fold enhanced transcription of CAM that is associated with a moderate (200–400 nM) elevation in cytoplasmic Ca^2+^, which is maintained for ~6 h via entry of extracellular Ca^2+^ due to SOCE-dependent opening of plasma membrane calcium channels (Figure 3A) [32,33]. During this first 6–10 h, TG-driven depletion of the ER Ca^2+^ pool results in an enhanced pool of unfolded proteins to which BiP chaperone binds, thus preventing its binding to a series of ER-membrane-bound stress sensors (Figure 3B). These include the RNA-activated protein kinase-like ER (PERK) kinase, the basic leucine-zipper activating transcription factor ATF6 (α and β), and the kinase endoribonuclease, IRE1. These stress sensors regulate components of the ER, which decreases protein synthesis as well as increases the ER’s capacity to fold the lowered level of newly synthesized proteins and to degrade misfolded/unfolded proteins (Figure 3B). Each of these stress sensors is negatively regulated by the chaperone protein BiP and, therefore, as misfolded proteins accumulate in the ER, they bind to BiP, disrupting its interaction with these ER stress sensors and activating UPR. Within the first 6 h of TG exposure, there is enhanced transcription of CaM and GADD 153 (also known as DNA Damage Inducible Transcript 3 (DDIT3) or CHOP; Figure 3C) and enhanced translation of a series of genes of UPR target genes (i.e., BiP; GADD 153; XBP1; Figure 3C,D) [14,32,33,34,35,36,37]. This is despite the fact that as part of this ER stress response, PERK is phosphorylated, which then phosphorylates eukaryotic translation initiation factor 2a (phosho-eIF2a; Figure 3D), attenuating translation of mRNA inhibiting protein synthesis (Figure 3E) [32,37]. By 24 h of TG treatment, this results in a decrease in GADD34 (also known as Protein Phosphatase 1, Regulatory Subunit 15A), Androgen Receptor (AR), Cyclin D_1_, and phospho-Rb protein (Figure 3D), and loss of Ki-67 expression, arresting the cells in G_0_ (Figure 3F).

Significantly, after 6 h of TG exposure and despite inhibition of protein synthesis, there is a doubling of CAM protein per cell (Figure 3E) [33]. CaM binding and regulation of the function of bound client proteins is dependent upon the extent of its binding Ca^2+^, which is low at 100 nM but saturated at 1 μM [38]. While the initial TG-induced elevation in cytoplasmic Ca^2+^ during the first 6 h is only from 50 to 200–400 nM and is, thus, not CaM saturating, this elevation is sufficient to enhance CaM binding to and, thus, upregulates the activation of the CaM client protein, the plasma membrane Ca^2+^-ATPase (PMCA-ATPase) pump. This is documented by the fact that microinjection of CaM inhibitory peptide inhibits this PMCA-ATPase pump and, thus, inhibits return of the Ca^2+^ to baseline (Figure 3G) after 6–12 h of TG exposure [37].

While the TG-induced depletion of the ER Ca^2+^ pool, induction of the UPR, and the initial cytoplasmic 200–400 nM Ca^2+^ transient elevation are necessary, they are not sufficient to irreversibly activate the DNA fragmentation and loss of clonogenic survival needed for the execution phase of apoptosis, which occurs >18 h later (Figure 3A and Figure 4A). This is documented by the fact that if after 3–6 h of TG exposure, during which depletion of the ER Ca^2+^ pool, induction of the UPR, and the initial cytoplasmic 200–400 nM Ca^2+^ transient elevation is completed, the cells are exposed to media lacking TG, no cell death occurs (i.e., no DNA fragmentation or loss of clonogenic survival; Figure 4A).

What is required to complete the initiation phase of TG-induced apoptosis is the occurrence of a CaM-dependent activation of a delayed (i.e., >18 h post-TG exposure) >10 μM elevation of cytoplasmic Ca^2+^ (Figure 3A) [15]. Under low nM Ca^2+^ conditions, the antiapoptotic Bcl-2 protein sequesters calcineurin. This is overcome by the initial Ca^2+^ influx within 6–12 h post TG exposure, which results in the subsequent phosphorylation of Bcl-2, preventing its sequestration of calcineurin. Subsequent to its release and binding Ca^2+^, calcineurin binds to the enhanced level of Ca^2+^-bound CaM protein (Figure 3E), displacing an auto-inhibitory site on calcineurin, which allows it to be active as a serine/threonine phosphatase. Once activated, calcineurin dephosphorylates a series of substrates. One of these targets is nuclear factor of activated T cells (NFAT). When dephosphorylated by calcineurin, cytosolic NFAT translocates to the nucleus by 18 h after treatment with TG [37]. Once in the nucleus, NFAT binds to DNA, interacting with other transcription factors, and enhances the transcription and translation of a series of genes including IP3 type 3 receptor (IP3R3) and Transient Receptor Potential Cation Channel Subfamily C1&3 (TRPC1&3; Figure 4B) [39]. The newly synthesized IP3R3 moves to the plasma membrane binding to TRPC1&3, forming a Receptor-Operated Ca^2+^ (ROC) channel, which allows extracellular Ca^2+^ to enter the cell [39,43]. This results in a second rise in cytoplasmic Ca^2+^ to a value of >10 μM occurring >18 h post return to baseline following the initial >500 nM rise in Ca^2+^ induced by TG exposure (Figure 3A), which is prevented by antisense constructs to IP3R3 [15]. The generation of the ROC channels marks the end of the initiation phase (Figure 5A), and beginning of the execution phase of TG-induced cell death (Figure 5B). This execution phase of TG-induced cell death is coincident with activation of caspase 3 and the flipping of phosphatidyl-serine from the intercellular to the extracellular side of the plasma membrane (Figure 4C). This is consistent with >10 μM elevation in cytoplasmic Ca^2+^ and caspase 3 proteolytic activity, which inhibits phosphatidyl-serine flippase while activating scramblase [44].

Additionally, coincident with this secondary >10 μM cytoplasmic Ca^2+^ rise is phosphorylation of Bcl-2, liberating it from calcineurin, which, under this high Ca^2+^ condition, like CaM, is now fully Ca^2+^ saturated. This results in maximum binding of CaM to Calcineurin, fully activating its ability to dephosphorylate the proapoptotic protein BAD. BAD, when phosphorylated, is sequestered in the cytoplasm bound to 14-3-3 (Figure 5B). Upon dephosphorylation by CaM-bound calcineurin, BAD translocates to the mitochondria, where it interacts with the apoptosis-related proteins Bcl-2 and Bcl-XL [37]. These proteins usually heterodimerize with the proapoptotic Bcl-2 family members, Bax and Bak. Once the Bax/Bak monomers are displaced from these heterodimers by BAD, they can bind to a series of other proteins, accelerating the opening of the mitochondrial permeability transition pore (MPTP), which involves a voltage-dependent anion channel (VDAC, also known as porin), resulting in the release of cytochrome C from the mitochondria into the cytoplasm (Figure 4D) [37].

Morphologically, the kinetics of this release of cytochrome C is coupled with TG-induced mitochondrial fragmentation, which occurs in two phases [41]. The initial fragmentation of mitochondria occurs rapidly within minutes of TG exposure and is dependent on an increase in cytoplasmic Ca^2+^ levels and Ca^2+^ influx into mitochondria. The initial mitochondrial fragmentation is a reversible transient (5 min–2 h) event, as tubular mitochondrial morphology is restored if the Ca^2+^ level decreases to baseline [41]. Prolonged incubation with TG induces a second phase after 30 h of mitochondrial fragmentation, which is non-reversible (Figure 4E) [41].

Once in the cytoplasm, cytochrome C binds to apoptosis protease activating factor (APAF-1), which complexes with procaspase 9. This complex formation results in the processing of inactive procaspase 9 to full enzymatic activity. Once activated, caspase 9 activates the processing of other procaspases, including procaspases 3 (Figure 4C), and 7 to active enzymes. The proteolytically active caspase then cleaves client proteins such as PARP (Figure 3D), and the 45-kD inhibitory subunit of the DNA fragmentation factor (DFF) complex, allowing release of the active 40 kD DNase subunit of DFF (Figure 5B). In addition, the delayed >10 μM cytoplasmic Ca^2+^ activates calpain-5 in mitochondria, which proteolyzes the 62 kD Apoptosis-Inducing Factor (AIF_62_), releasing the 57 kD truncated protein (AIF_57_) into the cytoplasm [45,46]. Once in the cytoplasm, AIF_57_ goes to the nucleus (Figure 4F). There, in combination with activated DFF, they fragment the genomic DNA (Figure 4G). It is this genomic fragmentation that irreversibly commits the cell to death and irreversible cellular fragmentation into apoptotic bodies (Figure 4H) [14,32,33,34,35,36,37].

## 5. Rationale for Development of Protease-Activated Thapsigargin Prodrug for Prostate Cancer Therapy

While TG is more than 100-fold more potent than ionomycin, it also can kill normal cells and, thus, there is no therapeutic index for the use of chemically unmodified TG clinically [40]. To produce a therapeutic index, John Isaacs realized that there are prostate differentiation restricted extracellular proteases that can be used to selectively hydrolyze and, thus, liberate cytotoxic cell killing moieties only at sites of metastatic prostate cancer, which spares normal host tissue toxicity. This approach was initially based upon the discovery by Hans Lilja at the University of Lund, Sweden, that Prostate-Specific Antigen (PSA), besides being a serum marker for prostate cancer detection, is a unique prostate differentiation restricted chymotrypsin-like serine protease whose enzymatic activity, once secreted by normal or malignant prostate epithelial cells into the serum, is inhibited by serum inhibitors [47,48,49]. Based upon these results, a collaboration was established between the Lilja laboratory at Lund and the Isaacs laboratory at Hopkins, with Sam Denmeade leading the Hopkins effort to identify specific peptide substrates whose efficient hydrolysis is restricted to PSA. This led to the identification of His-Ser-Ser-Lys-Leu-Gln-x-Leu (i.e., a 7AA peptide where -x- is the PSA cleavage site) as the lead peptide for such PSA-activated prodrug development [50,51]. As an in vitro proof of principle for such a PSA-activated prodrug approach, we collaborated with Andrew V. Schally, Department of Medicine, Tulane University School of Medicine and Veterans Affairs Medical Center, New Orleans, Louisiana to couple the primary amine of doxorubicin via a peptide bond to the carboxylic acid group of the 7AA lead peptide [51]. These studies documented that such a doxorubicin-peptide prodrug selectively kills cells that express enzymatically active PSA [52].

Doxorubicin prodrug development initially begun when Samuel Denmeade was a fellow and was continued as a collaboration when he became an independent faculty investigator at Hopkins. He subsequently documented that the PSA-peptide doxorubicin prodrug had no discernible systemic toxicity when given at four times the 100% lethal doxorubicin equivalent dose. An intraperitoneal (IP) dose of 60 mg/kg/week × 4 weeks resulted in a 57% decrease in human prostate cancer xenograft weight vs. control after 40 days and that a 25 mg/kg/week dose given by continuous infusion produced a similar decrease in tumor weight vs. control [53]. Based upon our preclinical validation of the approach, Merck Research Laboratories developed a different PSA-cleavable peptide-doxorubicin prodrug, which they took into clinical phase I trials [54]. Since Merck was working on a doxorubicin prodrug, we decided not to pursue our initial PSA-doxorubicin prodrug further but, instead, focus upon developing TG prodrugs.

The rationale for this decision is based upon the fact that TG’s killing ability requires entry into the cells in order to inhibit the SERCA pumps needed for cell killing. Thus, covalently coupling the water-soluble lead PSA-cleavable 7AA peptide via the carboxylic acid of its C-terminal leucine (L) to an amine-containing TG analog (Figure 1) produces a prodrug that is non-cell penetrant until hydrolyzed by PSA. When such a non-cell penetrant TG-prodrug is infused and is distributed systemically throughout the body via the blood, it remains inactive as a killing moiety, since circulating PSA is enzymatically inactive due to complex formation with serum protease inhibitors [55]. Once it diffuses into the extracellular fluid at metastatic sites of prostate cancer however, it can be efficiently hydrolyzed by the enzymatically active PSA restrictively present there, but not in normal tissue. Once hydrolyzed, the liberated amine-containing TG analog rapidly penetrates cells in its immediate vicinity due to its lipophilicity and does not re-enter the circulation, thus restricting its non-selective toxicity to the cancer microenvironment. One advantage of such selective extracellular hydrolysis is that only a fraction of the cancer cells need to express the prostate-restricted activating enzyme, since its continuous activity amplifies the level of amine-containing TG analog liberated into the extracellular fluid shared by all cells within the metastatic site. This extracellular amplification minimizes the problem of tumor cell heterogeneity by inducing a substantial “bystander effect”. This is because once the TG analog is liberated in the extracellular fluid within the tumor site, it kills both malignant and infiltrating host supportive cells—even those that do not express the activating enzyme. Thus, development of resistance due to tumor cell heterogeneity is retarded without simultaneously producing non-selective host toxicity [56].

## 6. Søren Brøgger Christensen and Identification of 12-ADT

To translate the protease-activated TG-prodrug strategy, in mid-1990, the Denmeade/Isaacs laboratories initiated a long-term highly productive collaboration with Søren Brøgger Christensen in order to identify the lead amine-containing TG analog for prodrug coupling [57,58,59,60,61,62,63,64,65]. This collaboration was based on the fact that Søren was the global expert on the isolation and medicinal chemistry of TG. Second, he was the main supplier of TG worldwide; the TG was purified from seeds of the *Thapsia garganica* plant which he, along with his wife Helle and their two daughters, personally harvested in July from the Iberian island of Ibiza, Spain. Once our collaboration started, initially, the Isaacs family (wife and three kids), then Sam Denmeade and his wife, joined the Christensen family in the July harvests on Ibiza (Figure 6).

Using the TG isolated from these harvested seeds, the Christensen laboratory developed synthetic approaches to systematically modify the side chains of the TG molecule in order to develop an analog containing a functional group that would allow coupling to a peptide. From these studies, it was determined that only the 2- and 8-position of TG molecules could be modified without seriously compromising their ability to inhibit the SERCA pump. Out of this extensive synthetic effort, a series of TG analogs were developed, in which the natural butanoyl-side chain in the O-8 position is replaced with side chain linkers, ending with a primary amine needed for peptide coupling [56,57]. This was based upon two facts. The first was the facile ability to chemically modify the O-8 position. The second was that TG binds the SERCA-2B pump at the interface of the ER membrane surface and the cytoplasmic region orienting the O-8 position side chains into the cytoplasmic region through a hydrophobic channel [66]. This means that if the O-8 position side chain is sufficiently long, then the positively charged primary amine at the end of the side chain will extend into the hydrophilic environment of the cytoplasm and not disrupt the lipophilic binding of TG [59]. On the basis of this insight, analogues of varying side chain length at the O-8 position with a charged moiety at the end were designed, synthesized, and then validated experimentally for their SERCA inhibition activity and cell killing. These results identified the 12-aminododecanoyl side chain analog of Thapsigargin (12ADT) as the lead amino-acid-containing TG analog that retained high lipophilicity as well as SERCA pump inhibition and, thus, cell killing for our TG prodrug approaches (Figure 1) [57,58,59,60,61,62,63,64,65].

## 7. First-Generation Thapsigargin Prodrugs

Once identified as the lead amine-containing TG analog, 12ADT was covalently coupled via the primary amine to the C-terminal carboxylic acid residue of the terminal leucine in our lead 7AA PSA peptide to produce our first-generation soluble, cell impermeable latent TG-prodrug (Figure 1) [58]. We documented that this L12ADT peptide prodrug is hydrolyzed efficiently by PSA, is selectively toxic to PSA-producing prostate cancer cells in vitro, and is stable in human plasma. The maximum tolerated multiday intravenous dose (MTD) of this PSA-TG prodrug is 7 mg/kg (4.4 μmoles/kg/dose), which results in a peak serum prodrug concentration of 15.4 ± 1.1 μM and an elimination half-life of ~2.8 h. After 24 h, less than 0.5% of free L12ADT is observed in the plasma. At this MTD, intratumoral concentration of the PSA-TG prodrug is 640 ± 80 nM (i.e., 8.5 times the LD_50_ for LNCaP cells in vitro) and 170 ± 58 nM for the liberated L12ADT (i.e., 13 times the LD_50_ for L12ADT in vitro).

Importantly, at this MTD of PSA-TG prodrug, growth of LNCaP prostate cancer xenografts in mice treated via SQ Alzet mini-pump for 4 weeks is completely inhibited without substantial host toxicity (Figure 7A), but has no effect on PSA non-producing renal carcinoma xenografts [58].

Based upon these preclinical results, a series of U.S. patents were awarded to the Isaacs/Denmeade/Christensen/Lilja co-inventors. To accelerate the clinical development of these TG prodrugs, Drs. Denmeade and Isaacs formed a collaboration with Dr. Craig Dionne, a corporate executive with extensive drug development experience, and together, they formed the start-up company GenSpera in 2003. As part of due diligence for fund raising, GenSpera was concerned with a potential issue regarding the broad claims in Merck patents for PSA-activated prodrugs. This led to a corporate decision to utilize additional prostate cancer-associated extracellular proteases for activation of 12-ADT-based prodrugs grounded on the portfolio of patents that had been previously assigned to GenSpera by the inventors.

## 8. Second-Generation Thapsigargin Prodrugs

There are at least three additional cancer-associated extracellular proteases as potential candidates for the activation of 12-ADT-based prodrugs. These are Human Glandular Kallikrein 2 (HK2 also known as KLK2), Fibroblast Activation Protein (FAP), and Prostate Specific Membrane Antigen (PSMA also known as FOLH1). HK2 is a trypsin-like protease, uniquely secreted into the extracellular fluid at high enzymatically active levels only by normal and malignant prostate epithelial cells [69,70]. Like PSA, once in the blood, its enzymatic activity is inhibited by serum protease inhibitors, making it an alternative candidate for prostate-targeted prodrug activation [69,70]. Another alternative candidate is the serine protease, FAP. This is based upon the studies of W. Nathaniel Brennen, initiated while a graduate student with Sam Denmeade, and then, as a post-doctoral fellow with John Isaacs. Subsequently, Dr. Brennen continued this collaboration when he became an independent faculty investigator at Hopkins. His studies focused on the tumor-promoting activity of the influx within sites of metastatic prostate cancer of tumor-infiltrating host-derived fibroblasts that have a highly increased plasma membrane expression of FAP [67,71,72,73,74]. FAP is a type II integral membrane serine prolyl protease of the dipeptidyl peptidase IV family, which is characterized by a unique post-prolyl cleavage specificity. Based on its restricted expression and unique substrate requirements, FAP is an ideal potential candidate for prodrug activation [67,71,72,73,74]. PSMA is highly expressed on the extracellular plasma membranes of prostate cancer cells and, as originally discovered by Warren D. W. Heston, has folate hydrolase enzymatic activity [75]. PSMA expression is also upregulated after ADT in resistant metastatic prostate cancer [76].

Based upon these findings, the Brennen/Denmeade/Isaacs group identified a series of highly restrictive peptide substrates uniquely recognized by HK2 [69], PSMA [64], or FAP [74], which, in collaboration with Søren Brøgger Christensen, were covalently coupled to 12ADT to produce prodrugs (Figure 1). Each of these prodrugs was tested and documented to be efficiently hydrolyzed only by its targeted protease. Each had anticancer efficacy in vitro vs. prostate cancer cell lines and in vivo vs. prostate cancer xenografts.

For the HK2-TG prodrug (Figure 1), the maximally tolerated multiday intravenous dose of prodrug is 6 mg/kg (3.67 μmoles/kg/dose), which produced a peak serum concentration of ~36 μM and has a half-life of ~40 min. In addition, over a 24 h period, <0.5% of free L12ADT analog is observed in plasma, while, within the cancer, the level of 12ADT-released toxin at this MTD is ~1 μM. The prodrug demonstrated a significant antitumor effect in vivo while being administered (Figure 7B), but prolonged intravenous administration is not possible due to local toxicity to tail veins [60].

For the FAP-TG prodrug (Figure 1), the maximally tolerated multiday intravenous dose of prodrug is 6.8 mg/kg (4 μmoles/kg/dose), which produced peak serum concentration of ~35–40 μM, and has a half-life of 4–6 h. In addition, over a 24 h period, <1% of free L12ADT analog is observed in plasma while, within the cancer, the level of S12ADT-released toxin at this MTD is 3–5 μM. The prodrug demonstrated a significant antitumor effect in vivo (Figure 7C). The antitumor effect is comparable to that observed with a maximally tolerated multiday intravenous dose of docetaxel, but results in significantly less toxicity [64].

For the PSMA-TG prodrug (Figure 1E), the maximally tolerated multiday intravenous dose of prodrug is 56 mg/kg (i.e., 40 μmole/kg/dose), which produced a peak serum concentration of ~800 μM and had a half-life of 4.9 h. In addition, over a 24 h period, <1% of free 12ADTβAsp analog is observed in plasma while, within the cancer, the level of released toxin at this MTD is >8 μM. The prodrug demonstrated a significant antitumor effect in vivo (Figure 7D). The antitumor effect is comparable to that observed with docetaxel, but results in significantly less toxicity [72].

The positive results with all four of the TG-based prodrugs (Figure 7) led to the issue of prioritization as to which of these prodrugs should first enter clinical development by GenSpera. This decision was ultimately decided based upon the fact that, first, the PMSA-TG prodrug had the highest multiday intravenous MTD, producing the highest level of selectively released toxin (i.e., 12ADTβAsp) within the cancers. Second, although initially thought to be prostate tissue-specific, PSMA is also expressed on the plasma membranes of tumor endothelial cells (ECs) in a large variety of solid malignancies, but not by ECs in normal tissues [77,78]. In addition, a radiolabeled humanized anti-PSMA antibody (J591) successfully targets metastatic sites in these additional malignancies. These findings suggested that one could target PSMA expression in the vasculature of most solid tumors to proteolytically activate a prodrug, leading to the selective death of cells within the site of metastases [79,80]. Thus, the therapeutic of such a PSMA-targeted TG prodrug would not be limited to only prostate cancer patients.

## 9. Mipsagargin (G202) as a PSMA-Activated Prodrug for Solid Malignancies

Based upon its expression by a large variety of solid malignancies, GenSpera chose our PSMA-activated prodrug to be the lead candidate for clinical development. GenSpera worked with locals on the island of Ibiza to establish the Spanish company Thapsibiza (https://thapsibiza.com (accessed on 4 December 2021)) to harvest large quantities of seeds from *T. garganica* and to begin larger scale cultivation of the plant. GenSpera further developed methods for safe, large-scale extraction of Thapsigargin from these seeds and developed GMP methods to scale up the synthesis (multi-gram to kg) of clinical grade Mipsagargin. Preclinical pharmacokinetic analysis (PK); toxicology; and chemistry, manufacturing, and controls (*CMC*) studies were completed to support filings required for an Investigational New Drug (*IND*) Application to the FDA, which gave its approval on September 4, 2009 (IND 105,830). Preclinical PK studies in mice, rats, and cynomolgus monkeys demonstrated virtually undetectable levels of the ultimate active PSMA cleavage product 12ADT-βAsp in plasma at any time after dosing and, hence, levels of this cleavage product were not assessed in clinical studies. A phase I trial was initiated to evaluate the safety, tolerability, and PK profile of intravenous (IV) administration of Mipsagargin in patients with locally advanced or metastatic solid tumors, refractory to standard therapy, or for whom no standard therapy was available [81]. These studies established the recommended phase II dosing (RP2D) regimen as IV infusion over a 1hr period of 40 mg m^−2^ of Mipsagargin on day 1, followed by infusion at the RP2D (66.8 mg/m^2^ or 1.58 mg/kg) on days 2 and 3 of each 28-day cycle with routine pre-medications and prophylactic hydration on each day of infusion. The dose-limiting toxicities of Mipsagargin included a hypersensitivity reaction that occurred on the first day of the infusion cycle and reversible elevation in serum creatinine that resolved before the end of a treatment cycle. At this infusion dose, Mipsagargin’s estimated half-life of tissue distribution (T_1/2-_α) is 2 h and its Volume of distribution at steady state (Vss) is 5 L/m^2^ (i.e., 0.135 L/kg) [80]. This Vss is essentially equivalent to the extracellular fluid volume (i.e., 7.3 L/m^2^ or 0.2 L/kg) in the human body, consistent with its inability to enter normal tissue. Its elimination phase half-life (T_1/2_ is 21 h [81].

While clinical response was not the end point for this phase 1 trial, prolonged disease stabilization was observed in a subset of patients, particularly those with metastatic hepatocellular carcinoma (HCC). Based upon these results, a Phase II multicenter, single-arm study was undertaken to evaluate the safety and efficacy of Mipsagargin in adult patients with advanced HCC who had progressed on or after treatment with sorafenib or were intolerant of sorafenib [82]. Assessments included time to disease progression (TTP), response rate, progression-free survival (PFS), overall survival (OS), and safety. The results documented that Mipsagargin is relatively well-tolerated and promotes prolonged disease stabilization in patients with advanced HCC that had progressed on prior treatment with sorafenib. The median TTP was 134.0 days, median PFS was 129.0 days, and median OS was 205.0 days [82]. The median observed TTP of 134.0 days is significantly greater than the historic control of approximately 63 days (*p* < 0.001) [82]. As part of the pharmacodynamics (PD) analysis, a subset of patients underwent dynamic contrast-enhanced magnetic resonance imaging (DCE-MRI) assessment of tumor blood flow metrics at baseline and within three days of completion of Mipsagargin administration in Cycle 2. This assessment documented a significant decrease in volume transfer coefficient (K_trans_) upon treatment, which suggested that Mipsagargin reduces blood flow in hepatic lesions consistent with the expression of PSMA by tumor endothelial cells within these lesions [82]. Based upon these combined results, the authors concluded that a larger clinical study to further characterize the activity of Mipsagargin in advanced HCC was warranted.

Unfortunately, this was not possible because in 2016, GenSpera was reorganized under new management, which excluded any of the original inventors. This reorganization also involved changing the name of the company to Inspyr Therapeutics. This company is now focused upon the development of a microparticle formulation of RT-AR001, an adenosine receptor antagonist for enhancing immune responses when administered intratumorally.

## 10. Future Approaches

Clinical development of TG prodrugs requires a continuous supply of TG, which, until recently, was isolated by Thapsibiza from harvested *T. garganica* seeds from Ibiza. This situation has changed, however. In a collaboration with Dr. Christensen, Drs. Tomáš Zimmermann, Pavel Drašar, and Michal Jurášek from the Department of Chemistry of Natural Compounds, University of Chemistry and Technology Prague, Czech Republic have isolated the structurally related sesquiterpene lactone, trilobolide, from cultivated wild type *L. trilobum*, which grows in the Czech Republic, and published a synthetic method for its conversion to 12-ADT [83].

With the cessation of GenSpera, the further clinical development of Thapsigargin-based prodrugs has been delayed. This delay provides an opportunity to integrate new discoveries, catalyzing a “re-thinking” of how to engineer better TG-based prodrugs and how optimally these should be used in combination with other drugs. For example, it has been documented that the potent killing ability of TG and the Leu-12ADT moiety liberated from the PSA, hK2, or FAP TG-based prodrugs involves not only their perturbation of ER, cytosolic, and mitochondrial Ca^2+^ dynamics leading to partial opening of the mitochondrial permeability transition pore, but also their inhibition of autophagy required for the loss of mitochondrial membrane potential needed for cytochrome C release [84]. In contrast, Asp-12ADT liberated from the PSMA TG-prodrug is reported to have a decreased ability to inhibit autophagy and, thus, combining it with inhibitors of autophagy increases the anticancer response [84]. This suggestion is consistent with our earlier report [68] that the therapeutic response of MCF-7 human breast cancer xenografts in estrogen-supplemented female mice to daily ×2 IV injections of 56 mg/kg/dose (i.e., 40 μmoles/kg/dose) Mipsagargin (G202) alone is able to produce significant (>50%) tumor regression. This regression is stabilized when combined with daily dosing with the oral HDAC4 inhibitor, Tasquinimod (TasQ; Figure 7E), which has been documented to inhibit autophagy [85]. These results warrant the clinical testing of a combination of daily oral TasQ with episodic IV dosing of Mipsagargin.

In these preclinical xenograft studies, Mipsagargin, even as a single agent, induces tumor regression when given at its MTD of 56 mg/kg/dose (i.e., 40 μmoles/kg/dose). In contrast, the PSA-, hK2-, and FAP-based TG prodrugs at their ten-fold lower MTD (i.e., ~4 μmoles/kg/dose) stop tumor growth, but do not produce tumor regression (Figure 7). Therefore, with regard to the further development of the PSA-, hK2-, or FAP-based TG prodrugs, additional chemical modifications are required. Modification to these peptide TG lipophilic prodrugs is needed to decrease non-specific cell penetration in normal tissue, thus increasing the MTD, and to increase their serum half-life of only a few hours following systemic dosing. To achieve this, Emmanuel Akinboye, as a postdoctoral fellow in the Denmeade/Isaacs group, developed a medicinal chemical approach to covalently couple a PSA-, hK2-, or FAP-cleavable peptide TG prodrugs via a stable maleimide-containing linker to cysteine-34 of human serum albumin (HSA; Figure 7F). This approach decreases dose-limiting host toxicity while enhancing plasma half-life from minutes to days (i.e., pharmacokinetic effect) and increases the tissue concentration of the maleimide-coupled albumin delivery (MAD) in the ECF at sites of cancer due to the enhanced permeability of albumin at these sites (i.e., enhanced permeability and retention effect) [86,87].

## Figures and Tables

**Figure 1 molecules-26-07469-f001:**
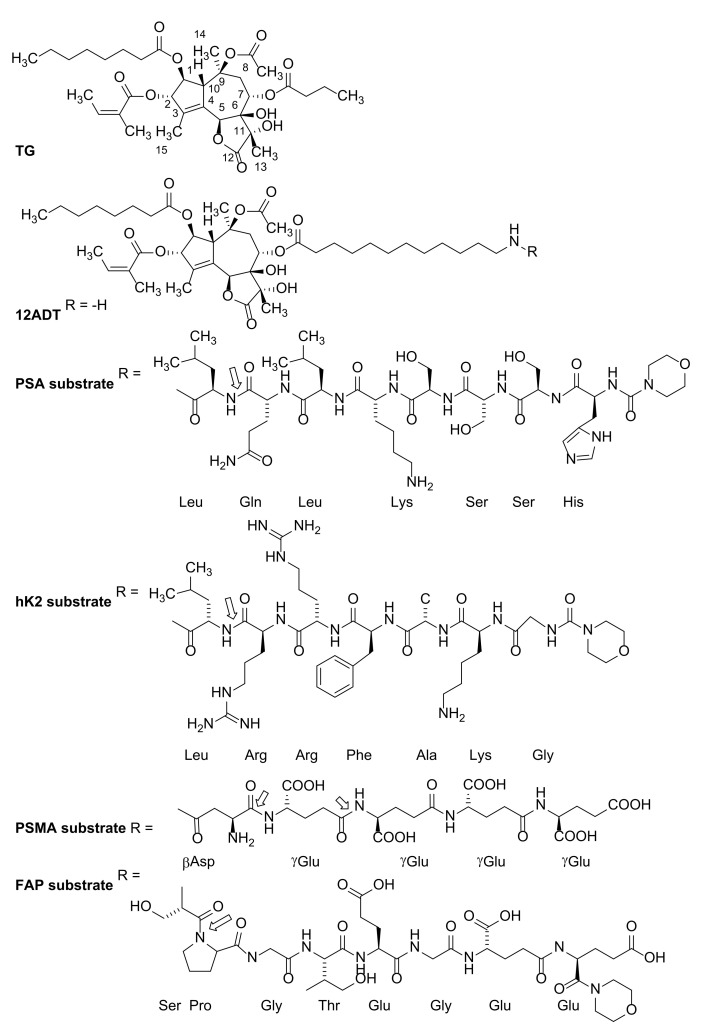
Structure for TG, its amine-containing 8-*O*-(12-aminododecanyl)-8-*O*-debutanoyl TG analogue [12ADT], and its peptide conjugates. Amino acid sequences used to restrict TG prodrug killing until hydrolyzed by: Prostate-Specific Antigen; hK2. Prostate Specific Membrane; PSMA and Antigen Fibroblast Activation Protein (FAP). Note: for the PSA, hK2, and FAP peptide prodrugs, the *N*-terminal amino group is capped with a morpholino-protection group. Open arrows denote sites of selective enzymatic hydrolysis.

**Figure 2 molecules-26-07469-f002:**
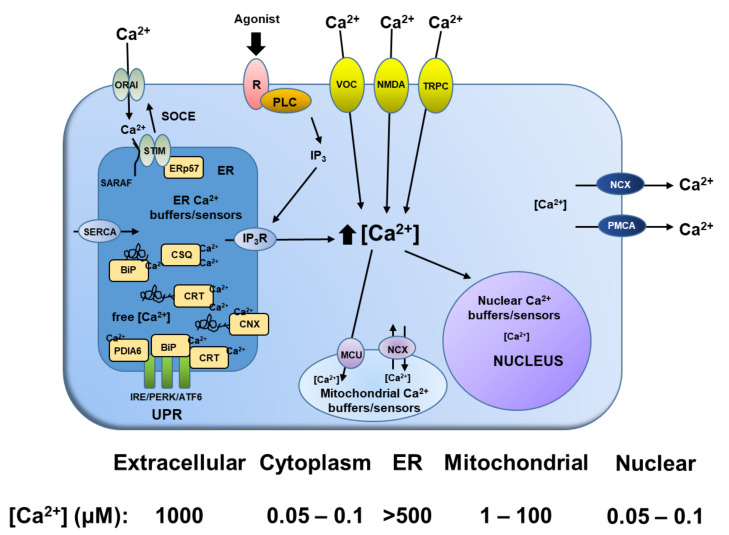
Overview of compartmental calcium gradients and their function in mammalian cells (Redrawn from reference [26]. Abbreviations: ER (Endoplasmic Reticulum); SERCA (Sarco/endoplasmic reticulum calcium ATPase); PMCA (Plasma membrane calcium ATPase); R (Receptor); PLC (Phospholipase C); IP3 (inositol-1,4,5-trisphosphate); IP3R (Inositol-1,4,5-trisphosphate receptor); BiP (Binding-Immunoglobulin Protein also known as Heat Shock 70 kDa Protein 5 or Glucose-Regulated Protein, 78 kDa); CRT (Calreticulin); CNX (Calnexin); PDIA6 (Protein Disulfide Isomerase Family A Member 6); CSQ (Calsequestrin); ERp57 (ER protein 57 also known as Protein Disulfide Isomerase Family A Member 3); STIM (Stromal Interaction Molecule 1); SARAF (Store-Operated Calcium Entry Associated Regulatory Factor); SOCE (Store-operated Ca^2+^ entry); Orai1 (ORAI Calcium Release-Activated Calcium Modulator 1); UPR (Unfolded Protein Response); IRE (Inositol-Requiring Enzyme also known as ERN1); PERK (PRKR-Like Endoplasmic Reticulum Kinase also known as EIF2AK3); ATF6 (Activating Transcription Factor 6); Voltage-gated Ca^2+^ channel (VOC); Transient receptor potential channel (TRPC); Na^+^/Ca^2+^ exchanger (NCX); and Mitochondrial Ca^2+^ uniporter (MCU).

**Figure 3 molecules-26-07469-f003:**
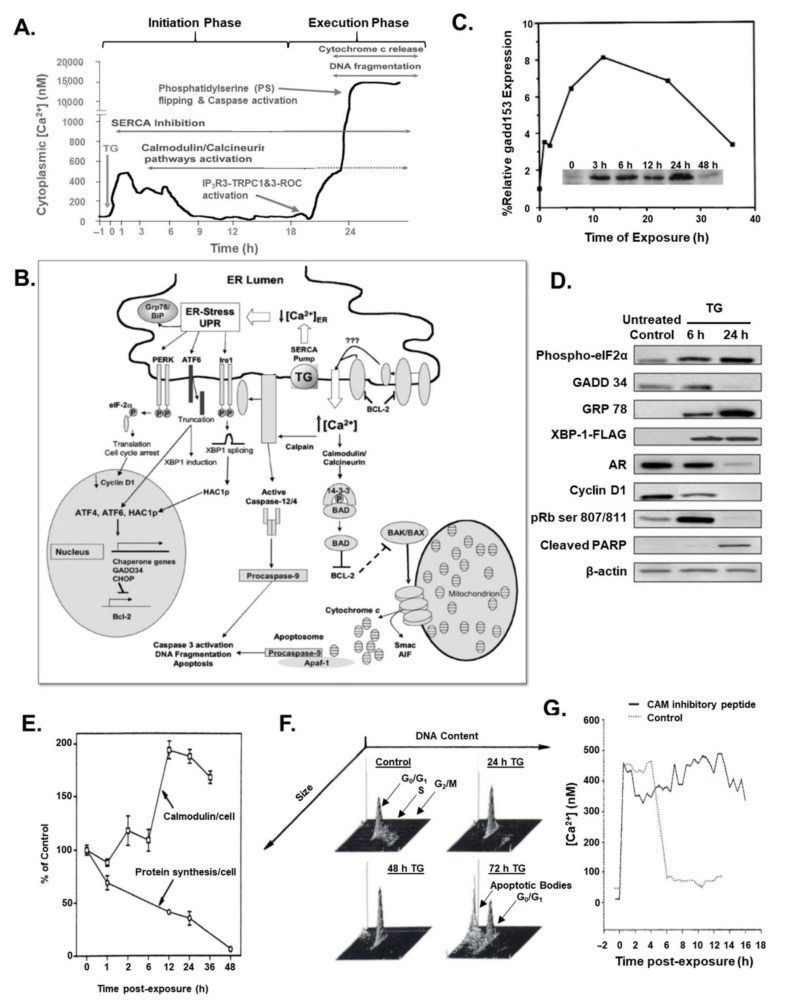
(**A**) Time course of changes in cytoplasmic Ca^2+^ vs. indicated biochemical parameter following exposure of prostate cancer cells to 500 nM Thapsigargin from reference [35]. (**B**) Overview of the apoptotic pathways induced by TG inhibition of SERCA pump involving depletion of ER Ca^2+^, elevation of cytoplasmic Ca^2+^, and the activation of calmodulin/calcineurin/calpain cytoplasmic-mitochondrial-dependent caspase/nuclease activation and irreversible DNA fragmentation from reference [35]. See text for details. (**C**–**F**) Kinetic changes following treatment of prostate cancer cells with 500 nM TG normalized on a per cell basis in: (**C**) Gadd153 (also known as BiP) mRNA denoted with the line from reference [37] and the protein level in the insert from reference [15]; (**D**) Indicated proteins from reference [38]; (**E**) Protein synthesis and calmodulin protein from reference [32]; and (**F**) Cell cycle progression based upon flow cytometer determined DNA content vs. cell size (Small arrows indicate position of G_0_/G_1_ vs. S vs. G_2_/Mitosis) from reference [32]; (**G**) Cytoplasmic Ca^2+^ in control vs. CAM inhibitory peptide microinjected cells from reference [37].

**Figure 4 molecules-26-07469-f004:**
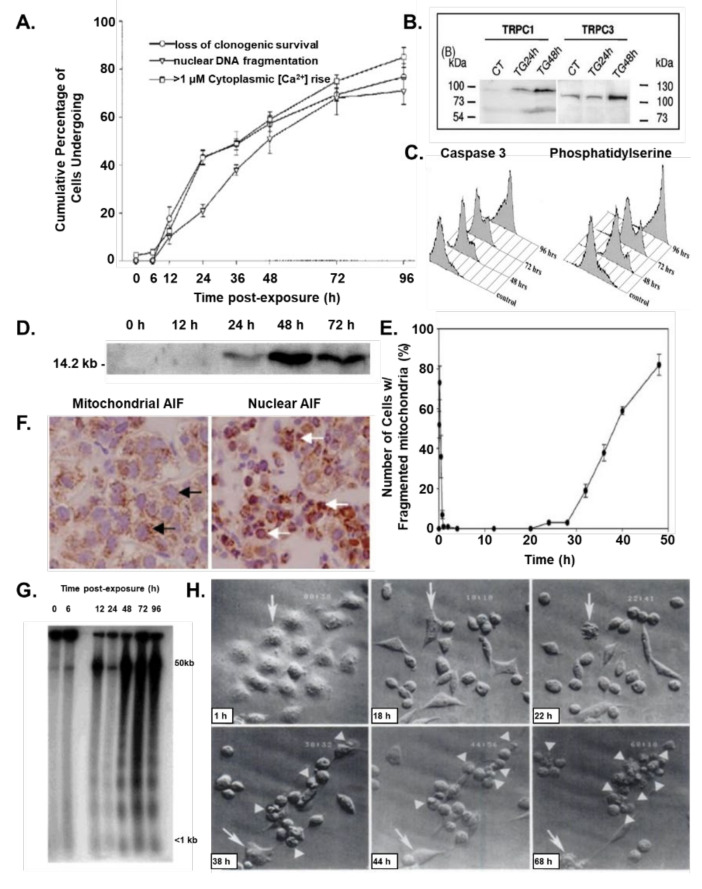
Kinetic changes following treatment of prostate cancer cells to 500 nM Thapsigargin in: (**A**) μM rise in cytoplasmic Ca^2+^ vs. loss of clonogenic survival vs. irreversible nuclear DNA fragmentation from reference [37]; (**B**) Upregulation of TRPC1 and 3 protein from reference [39]; (**C**) Percentage of cells expressing activated caspase 3 intracellularly detected based upon a cell penetrant fluorescence substrate and expressing phosphatidyl-serine extracellularly detected based upon plasma membrane binding of fluorescently labeled annexin V from reference [40]; (**D**) Cytochrome C protein release from mitochondria to cytoplasm from reference [37]; (**D**) Percentage of cells with mitochondrial fragmentation from reference [41]; (**E**) Translocation of AIF from mitochondria (black arrows) at time zero vs. to cell nucleus (white arrows) at 48 h post-TG exposure [42]; (**F**) Cell counting for mitochondrial fragmentation in prolonged treatment TG showed the second phase fragmentation of mitochondria after a 32 h incubation [41]; (**G**) Irreversible fragmentation of genomic DNA from reference [37]; and (**H**) Cellular fragmentation into apoptotic bodies with arrow and arrow heads indicating same cell followed longitudinally from reference [32].

**Figure 5 molecules-26-07469-f005:**
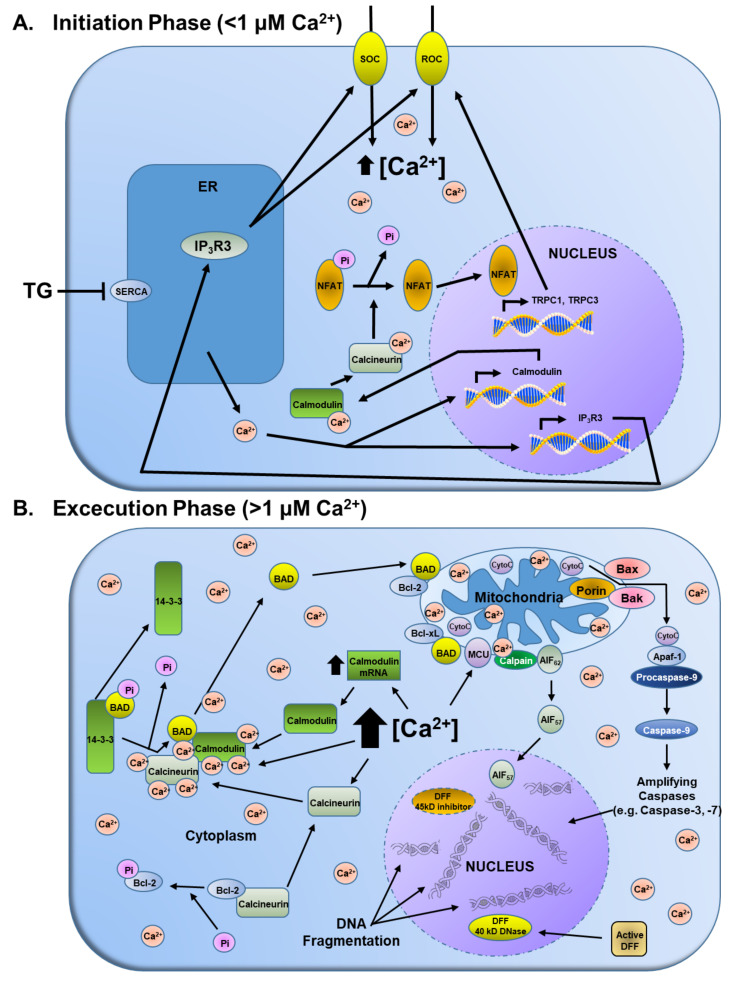
Overview of the downstream signaling pathways in the (**A**) <1 μM Ca^2+^ initiation phase vs. (**B**) >1 μM execution phase of prostate cancer cell death induced by Thapsigargin.

**Figure 6 molecules-26-07469-f006:**
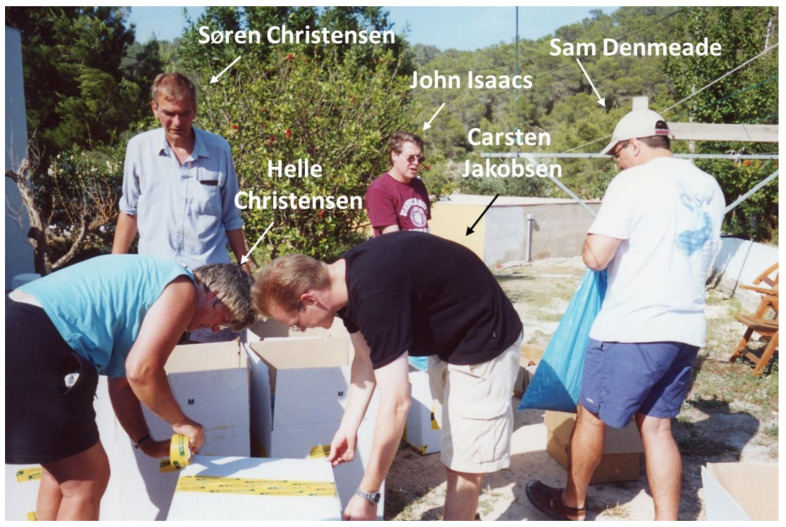
Photograph of Soren Christensen and his wife Helle, along with Carston Jakobsen, a postdoctoral fellow from the Christensen laboratory, and Samuel Denmeade and John Isaacs boxing up the >100 kg of *Thapsia garganica* seeds harvested from Ibiza, Spain in the first week of July.

**Figure 7 molecules-26-07469-f007:**
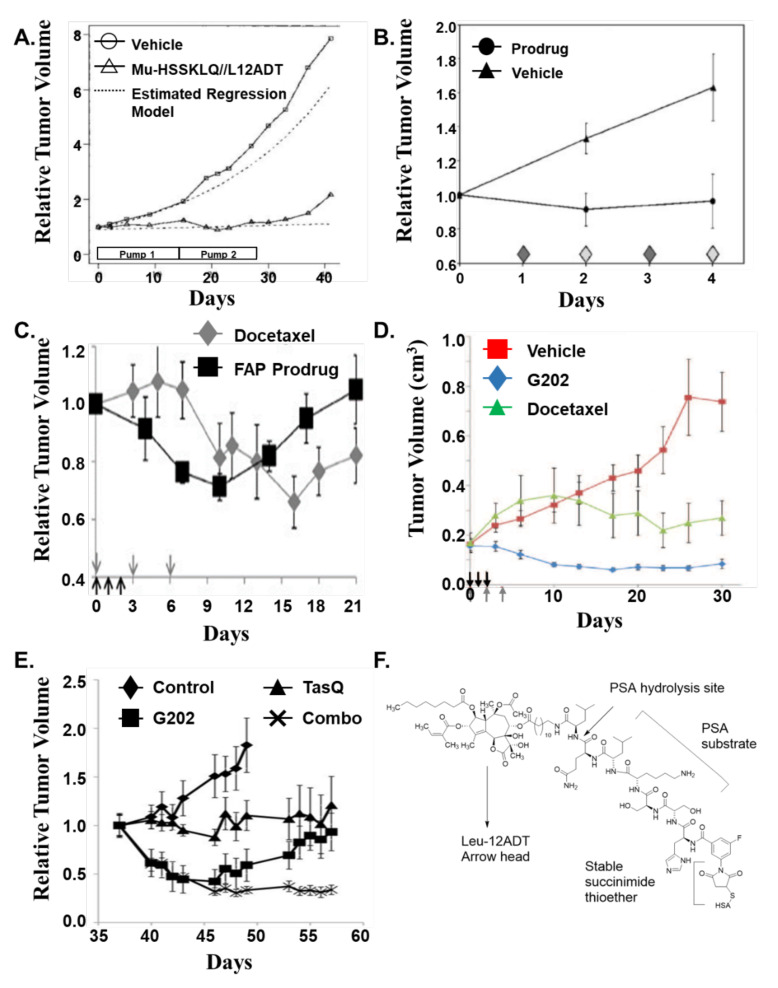
In vivo response of LNCaP human prostate cancer xenografts in intact male immune-deficient mice to: (**A**) PSA-TG prodrug given via SQ alzet mini-pump at 25 mg/kg/week (i.e., 2.5 μmoles/kg/day) for 4 weeks from reference [45]; (**B**) hK2-TG prodrug given IV at 6 mg/kg/injection (i.e., 3.67 μmoles/kg/dose) once a day for 4 consecutive days from reference [60]; (**C**) FAP-TG prodrug given i.v. at 6.8 mg/kg (4 μmoles/kg/dose) once a day for 3 injections vs. docetaxel given i.v. at 0.39 μmoles on every 3 days for 3 injections from reference [67]; (**D**) PSMA-TG prodrug (also known as G202) given i.v. at 56 mg/kg (i.e., 40 μmoles/kg/dose) once a day for 3 injections vs. docetaxel given i.v. at 0.39 μmoles on every 3 days for 3 injections from reference [64]. (**E**) Response of established (0.8 cc) MCF-7 human breast cancers growing in mice given 2 daily intravenous injections at 56 mg/kg of Mipsagargin prodrug (G202) alone and in combination with 10 mg/kg/d oral Tasquinimod (TasQ). Results are presented as relative tumor size normalized to tumor volume at initiation of treatment. *p* < 0.05 for combination (combo) group vs. monotherapies after day 49 [68]. (**F**) Structure of prostate-specific antigen (PSA)-activated prodrug of L-Leu-12ADT with 2-fluoro-5-maleimidobenzamide covalently bound to the N-terminal of PSA substrate histidine-serine-serine-lysine-leucine-glutamic acid (HSSKLQ) for coupling to the cysteine 34 of human serum albumin.

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
