# Peer review of "Mipsagargin: The Beginning—Not the End—of Thapsigargin Prodrug-Based Cancer Therapeutics"

_molecules, 2021, doi:10.3390/molecules26247469_

Round 1
Reviewer 1 Report
This review discusses the history of the cell-penetrant sesquiterpene-lactone thapsigargin and its prodrug analogue Mipsigargin. The text is well written and there is excellent use of figures and illustrations throughout. The chronology of the review is engaging. The authors are to be congratulated for an excellent summation of 40 years of history. Søren Christensen is to be commended for his insight and lifelong persistence. These are the traits of a brilliant scientist.
General comments:
Section 1 is a brief but adequate summary of prostate cancer therapeutics.
Ln 39; LHRH – define abbreviation on first usage (luteinizing hormone-releasing hormone).
Section 2 provides a clear well written summary of the rationale underlying the search for intracellular Ca2+ disruptors.
Section 3 describes compartmental calcium gradients, providing useful context for readers.
Ln 122; underlining of text seems unnecessary/inappropriate
Section 4 introduces the chronological utilisation of Thapsigargin by various laboratories coupled with an elegant and detailed description of the underlying mechanisms that were eventually revealed.
Section 5 introduces the rationale for development of protease activated Thapsigargin analogues using protease cleavable peptides to generate a prodrug which is non-cell penetrant until specifically hydrolyzed by PSA.
Ln 416 – "a.k.a." is unnecessary. Just state abbreviation i.e. (PSA)
Section 6 and 7 describes the structure-based evaluation of Thapsigargin analogues to identify a lead amino-acid containing TG candidate from which a latent TG-prodrug was derived, termed ‘PSA-TG prodrug’. In vitro and in vivo experiments are described that provided initial validation of therapeutic selectivity.
Figure 7F. Consider avoiding the unusual perspective – planer 2D image is preferred
Section 8 describes the 2nd Gen prodrug design – a series of discrete prodrugs found to be efficiently hydrolyzed by specific target proteases, with positive results reported for all 4 of the TG-based prodrugs.
Ln 602 – ‘that that’ – remove repetition
Section 9 captures the decisions and process optimisation necessary for clinical evaluation. A concise but informative summary of the Phase 1 and 2 findings are provided.
Author Response
LHRH has been defined
Figure 2 has been changed
A.k.a. has been removed.
Figure 7 has been changed
Repletion has been removed
Reviewer 2 Report
This manuscript does an excellent job demonstrating the preclinical validation of each of these 31 TG-prodrugs with special attention to the PSMA TG-prodrug. This is an exceptionally good review and comprehensively introduced the thapsigargin drug development.
Author Response
Fig.1 The thapsigargin skeleton has been numbered.
A conclusion has been added (section 10)
LHRH has been explained
Figure 7 has been revised
The underline has been removed
Leucine has been changed to leucine
Reviewer 3 Report
I have carefully read the manuscript of a review entitled “Mipsigargin: The Beginning not the End of Thapsigargin Pro- 2 drug Based Cancer Therapeutics”. I should note that the subject of this review, with its focus on natural compounds in medicine-inspired chemistry, lies somewhat aside from my scientific field. However, I have read the manuscript with great interest and can say that it undoubtedly deserves publication, however, after certain improvements.
First of all, the Introduction section is overloaded with abbreviations, but does not provide logical connection between the subject of research and the field of prostate cancer. The next point is a lack of Conclusion. I miss a short essay which briefly summarizes the main steps in thapsigargin story, for example something like “…thapsigargin is a potent cytotoxin inducing apoptosis upon inhibition of SERCA pump required for cellular viability… Thapsigargin has been thoroughly studied as an antineoplastic agent… However, its high toxicity to normal cells significantly hampered its application. This obstacle was tried to overcome by...” and so far, including very brief state-of-the-art. Or has the story - indeed - ended in 2016? The home page of Inspyr therapeutics shows thatmipsagargin is still in the Phase II trials (as for February 2020). The latest review I found on the topic is of the year 2020 ( Jaskulska A, Janecka AE, Gach-Janczak K. Int J Mol Sci. 2020). This review should be cited.
Figure 1: It would be helpful if the authors numerate the atoms within the structure of parent TG and – optionally - mark pharmacophores essential for the biologic properties as well as the sites for modifications. Indeed, 11 pages later they discuss that only “the 2- and 8-position of TG molecule could be modified without seriously compromising the ability to inhibit the SERCA pump” and I would recommend to make here a reference to Figure 1.
Figure 7F: Something went wrong with a formatting of the chemical structure and it is stretched vertically. Should be redone.
Line 39: LHRH – expand abbreviation
Line 122: “from the seeds of the fruits of the” - what should this underline mean?
Line 446: “l” in “leucine” should be small
Author Response
Leucine had been changed to leucine
Figure 7 has been changed